# Dynamic proton-dependent motors power type IX secretion and gliding motility in *Flavobacterium*

**Maxence S. Vincent**[1¤], **Caterina Comas Hervada**[1], **Corinne Sebban-Kreuzer**[1], **Hugo Le Guenno**[2], **Maïalène Chabalier**[1], **Artemis Kosta**[2], **Françoise Guerlesquin**[1], **Tâm Mignot**[3], **Mark J. McBride**[4], **Eric Cascales**[1]*, **Thierry Doan**[1]*

1 Laboratoire d'Ingénierie des Systèmes Macromoléculaires, Institut de Microbiologie, Bioénergies et Biotechnologie, Aix-Marseille Université – CNRS UMR7255, Marseille, France, 2 Microscopy Core Facility, Institut de Microbiologie, Bioénergies et Biotechnologie, Aix-Marseille Université, Marseille, France, 3 Laboratoire de Chimie Bactérienne, Institut de Microbiologie, Bioénergies et Biotechnologie, Aix-Marseille Université – CNRS UMR7283, Marseille, France, 4 Department of Biological Sciences, University of Wisconsin-Milwaukee, Milwaukee, Wisconsin, United States of America

¤ Current address: Department of Biochemistry, University of Oxford, Oxford, United Kingdom
* cascales@imm.cnrs.fr (EC); tdoan@imm.cnrs.fr (TD)

**Data Availability Statement:** All relevant data are within the paper and its Supporting Information files.

## Abstract

Motile bacteria usually rely on external apparatus like flagella for swimming or pili for twitching. By contrast, gliding bacteria do not rely on obvious surface appendages to move on solid surfaces. *Flavobacterium johnsoniae* and other bacteria in the Bacteroidetes phylum use adhesins whose movement on the cell surface supports motility. In *F. johnsoniae*, secretion and helicoidal motion of the main adhesin SprB are intimately linked and depend on the type IX secretion system (T9SS). Both processes necessitate the proton motive force (PMF), which is thought to fuel a molecular motor that comprises the GldL and GldM cytoplasmic membrane proteins. Here, we show that *F. johnsoniae* gliding motility is powered by the pH gradient component of the PMF. We further delineate the interaction network between the GldLM transmembrane helices (TMHs) and show that conserved glutamate residues in GldL TMH2 are essential for gliding motility, although having distinct roles in SprB secretion and motion. We then demonstrate that the PMF and GldL trigger conformational changes in the GldM periplasmic domain. We finally show that multiple GldLM complexes are distributed in the membrane, suggesting that a network of motors may be present to move SprB along a helical path on the cell surface. Altogether, our results provide evidence that GldL and GldM assemble dynamic membrane channels that use the proton gradient to power both T9SS-dependent secretion of SprB and its motion at the cell surface.

## Introduction

*Flavobacterium johnsoniae*, one of the fastest gliding bacteria described to date, uses surface-anchored adhesins to move on solid surfaces [1–3]. Remarkably, the major adhesin SprB exhibits a rotational behavior [4], and its motion at the cell surface describes a closed helicoidal

**Funding:** This work was supported by the Aix-Marseille Université (https://www.univ-amu.fr/), the Centre National de la Recherche Scientifique (https://www.cnrs.fr/fr/page-daccueil), and grants from the Agence Nationale de la Recherche (ANR-15-CE11-0039 and ANR-20-CE11-0017), from the Excellence Initiative of Aix-Marseille University (A*MIDEX, A-M-AAP-ID-17-33-170301-07.22), a French "Investissements d'Avenir" programme, and from the Fondation Bettencourt-Schueller (https://www.fondationbs.org/fr/sciences-de-la-vie). M.S.V. was supported by a doctoral fellowship from the French Ministère de la Recherche, and an end-of-thesis fellowship from the Fondation pour la Recherche Médicale (FDT2018-05005242). The funders had no role in study design, data collection and analysis, decision to publish, or preparation of the manuscript.

**Competing interests:** The authors have declared that no competing interests exist.

**Abbreviations:** CCCP, carbonyl cyanide m-chlorophenyl hydrazine; cryo-EM, cryo-electron microscopy; CYE, Casitone Yeast Extract; DCCD, N-N'-Dicyclohexyl-carbodimide; DPC, dodecylphosphorylcholine; FRAP, fluorescence recovery after photobleaching; IM, inner membrane; IPTG, iso-propyl-β-D-thio-galactopyranoside; MM, Motility Medium; NMR, nuclear magnetic resonance; OM, outer membrane; PMF, proton motive force; SIM, structured illumination microscopy; T9SS, type IX secretion system; TMHs, transmembrane helices.

pattern along the long axis of the cell [5,6]. It is proposed that binding of SprB to the substratum generates adhesion points and hence that SprB motion relative to the cell displaces the cell body in a forward screw-like motion [5,7]. SprB and other adhesins involved in gliding motility are transported to the cell surface by a multiprotein secretion apparatus, named type IX secretion system (T9SS) [8–10], which is present in most bacteria in the Bacteroidetes phylum [11,12].

The T9SS was discovered in the opportunistic pathogen *Porphyromonas gingivalis* in which it conveys a large number of virulence factors, including gingipain proteinases, across the outer membrane (OM) to the cell surface or the extracellular milieu [13–15]. In addition to gliding adhesins and gingipain proteinases, the repertoire of T9SS substrates also includes enzymes involved in nutrient supply and biofilm formation [13,16–18]. While the roles of the T9SSs and their substrates are relatively well known, information on T9SS architecture and mechanism of action are still sparse. However, conserved features of the T9SS have recently emerged [19–22]. The common T9SS architecture includes (i) the trans-envelope complex GldKLMN composed of 2 inner membrane (IM) proteins, GldL (or PorL) and GldM (or PorM), and of an OM-associated ring complex composed of the GldK OM lipoprotein and the GldN periplasmic protein [23–25]; (ii) the SprA (or Sov) OM translocon [26]; and (iii) the attachment complex that is comprised of the PorU, PorV, and PorZ proteins [27–31]. These proteins assemble through a dense network of interactions that are poorly characterized and likely involve other conserved T9SS subunits.

T9SS-dependent secretion and gliding motility is a process energized by the IM proton motive force (PMF) because inhibitors that dissipate the PMF prevent substrate secretion and halt cell displacement [32–35]. At the single-cell level, Nakane and colleagues directly observed that SprB dynamics halted almost immediately after the addition of carbonyl cyanide m-chlorophenyl hydrazone (CCCP), a protonophore that collapses the PMF [6]. Hence, it was proposed that a PMF-dependent motor powers SprB dynamics and cell gliding. The nature of the molecular motor that powers SprB motion has been a long-standing question. Among the T9SS core components, only the GldL and GldM IM proteins share features with recognized PMF-dependent motors involved in the energization of flagellum rotation (MotAB), iron acquisition (ExbBD), OM stability (TolQR), or myxococcal gliding motility and sporulation (AglQRS) [36–41]. GldL presents 2 transmembrane helices (TMHs) and largely faces the cytoplasm, while GldM is a bitopic protein with a large periplasmic carboxyl-terminal domain. The *P. gingivalis* homologs of GldL and GldM (PorL and PorM, respectively) interact via their TMHs [23]. In addition, structural studies showed that the GldM and PorM periplasmic regions form dimers and are composed of 4 domains, from D1 to D4 [25]. The carboxyl-terminal D4 domain of PorM is involved in interactions with the OM-associated PorKN complex [25]. Finally, GldL/PorL and GldM/PorM bear conserved glutamate residues that may participate in harvesting the PMF [20,23]. GldL and GldM are thus ideal candidates for constituting the IM proton-dependent motor powering type IX secretion and/or SprB dynamics. Indeed, a recent study presented the cryo-electron microscopy (cryo-EM) structure of the GldLM complex [42]. The complex comprises 2 single TMHs of a GldM dimer inside a pentameric ring of GldL, an architecture common with other known motors. The study also provided evidence that inter-TMH contacts modulated by the PMF are important for motor function. In addition, protonatable residues located in the GldL TMHs were shown to be essential for motor function [42]. Here, we provide further support and expand these conclusions. We show that the proton gradient component of the PMF is the source of energy powering gliding motility. We further demonstrate that the function of the GldLM motor requires a highly conserved glutamate residue in GldL, E49, whose protonation state controls interactions between the GldL and GldM TMHs and GldM conformation. We then show that substitution of a second

GldL glutamate residue, E59, had no effect on secretion of SprB to the cell surface, but abolished SprB movement, thereby constituting a tool to uncouple T9SS-dependent secretion and gliding motility. Based on these results, we propose an updated model in which GldM conformational change upon PMF sensing is transmitted into mechanical torque through the periplasmic part of the T9SS to drive SprB motion.

## Results and discussion

### Gliding is energized by the proton gradient

It is well known that gliding motility is arrested upon dissipation of the PMF [31–35]. The PMF consists of 2 gradients across the cytoplasmic membrane: an electrical potential ($\Delta\psi$) and a chemical potential ($\Delta$pH). To better define the energy source that powers gliding motility, *F. johnsoniae* gliding cells in a well chamber with glass bottom were subjected to valinomycin/K⁺ or nigericin, to specifically dissipate $\Delta\psi$ or $\Delta$pH, respectively, and single-cell gliding motility was quantified (Fig 1A and 1B). In agreement with previous observations [6], cells glide with

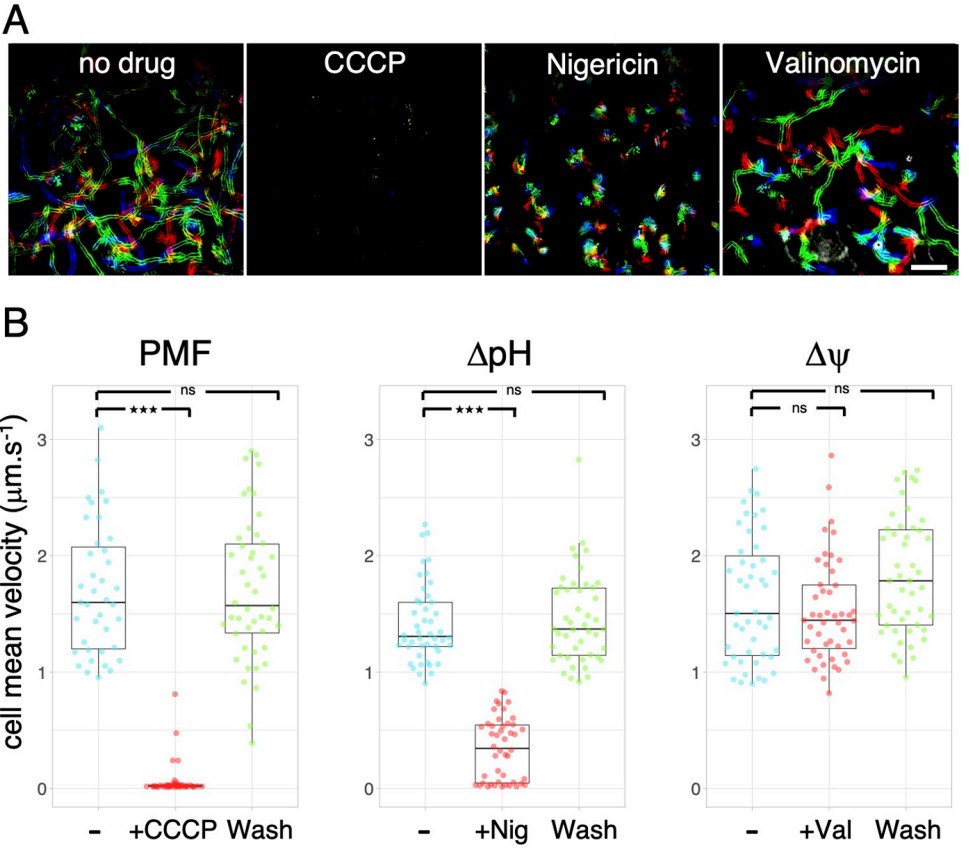

**Fig 1. Effects of PMF dissipating drugs on *F. johnsoniae* cells gliding. (A)** Rainbow traces of cell motility on glass recorded by phase contrast microscopy over time (2 minutes) in the absence of drug or in the presence of CCCP, nigericin, or valinomycin. Individual frames from time-lapse acquisition were colored from red (start) to yellow, green, cyan, and blue (end) and merged into a single rainbow image. Scale bar, 20 μm. **(B)** Combined jitter plots/boxplots of mean cell gliding velocity (in μm.s⁻¹) of *n* > 38 WT cells before (−), during a pulse of 10 μM CCCP (+CCCP), or 7 μM nigericin (+Nig), or 40 μM valinomycin/+50mM KCl (+Val), and after wash with fresh CYE medium (Wash). Statistical significance relative to the nontreated condition (−) is indicated above the plots (ns, nonsignificative; ***, *p* < 0.001; Wilcoxon *t* test). Data underlying this figure can be found in S1 Data. CCCP, carbonyl cyanide m-chlorophenyl hydrazone; PMF, proton motive force; WT, wild-type.

an average speed of 1.7 µm.s$^{-1}$ (Fig 1B). As a control, injection of 10 µM of CCCP rapidly blocked all cell displacement in a reversible manner. By contrast, no significant inhibitory effect was observed upon addition of 40 µM valinomycin (+50 mM KCl). However, when cells were treated with 7 µM nigericin, motility was strongly impaired (Fig 1B). Instead of gliding, cells appeared to jiggle around the same location, possibly because nigericin did not totally abolish the ΔpH (Fig 1A). When nigericin was washed out, cells resumed gliding motility at normal speed. Therefore, we conclude that the proton gradient, but not the electrical potential, is the source of energy used by the gliding machinery.

## GldL and GldM constitute the molecular motor that couples PMF to GldM conformational changes

Bacterial molecular motors such as the MotAB flagellar stator, the ExbBD and TolQR related transport systems, and the AglRQS gliding motor generate mechanical energy by harvesting the chemical gradient through the cytoplasmic membrane [40,43–45]. These complexes usually comprise 2 subunits organized in a 5:2 stoichiometry that interact via their TMHs [46,47]. A conserved acidic residue, located in one TMH and facing the other TMHs, plays a key role in proton transit [46–50]. It is proposed that proton flow through the channel triggers protonation–deprotonation cycles of the side chain of this residue and induces rearrangements in the TMHs, ultimately leading to the production of mechanical torque in the form of conformational changes in the extramembrane regions [51–53].

**GldL and GldM interact via their transmembrane segments.** In agreement with the recent cryo-EM structure of the GldLM complex, bacterial 2-hybrid analyses show that GldL and GldM interact (Fig 2A). As previously shown for the *P. gingivalis* PorLM complex [23], these interactions likely involve the transmembrane segments of both proteins. In agreement with the cellular localization of the soluble domains of GldL and GldM, which reside in the cytoplasm and periplasm, respectively, no GldL-M interaction was observed when only the soluble domains of these proteins (GldL$_C$ and GldM$_P$) were tested (Fig 2A). Finally, similarly to the *P. gingivalis* PorKLMN complex [23,24], the GldLM module is implicated in interactions with the putative OM-associated GldKN/O ring (S1A Fig) via contacts between the GldM periplasmic domain and GldK, GldN, and GldO (S1B Fig). To test the contribution of TMHs to GldLM interactions, we conducted GALLEX and BLA approaches. GALLEX is based on the repression of a β-galactosidase reporter by 2 LexA DNA binding domains with different DNA binding specificities (LexA$^{WT}$ and LexA$^{408}$). If 2 TMHs interact, LexA$^{WT}$ and LexA$^{408}$ heterodimeric association causes repression of β-galactosidase synthesis [54,55]. We tested interactions between TMHs displaying in-to-out topologies (GldL-TMH1 and GldM-TMH; Fig 2B). GldL-TMH1 and GldM-TMH specifically formed homodimers but no interaction was detected between GldL-TMH1 and GldM TMH (Fig 2C). To test the interaction with GldL-TMH2, which exhibits an out-to-in topology, we used the BLA assay [56,57]. GldL-TMHs and GldM-TMH were fused to either the N- or carboxyl-terminal domain of β-lactamase (Bla). If an interaction between TMHs occurs, a functional β-lactamase is reconstituted, and its activity can be quantified using a chromogenic substrate-based assay. In this assay, GldL-TMH2 homodimerization was not observed. However, GldL-TMH2 specifically interacted with both GldL-TMH1 and GldM-TMH (Fig 2D). Taken together, our data show that GldL TMH1 and TMH2 interact with each other, and GldL TMH2 interacts with GldM single TMH in the motor complex. With the exception of GldL-TMH1/GldL-TMH1 contacts detected by GALLEX, these data are in agreement with the position of the TMHs in the recent cryo-EM structure of the GldLM complex [42].

**GldM changes conformation depending on the proton gradient.** We next sought to understand how the GldLM complex responds to the proton gradient. Our previous structural

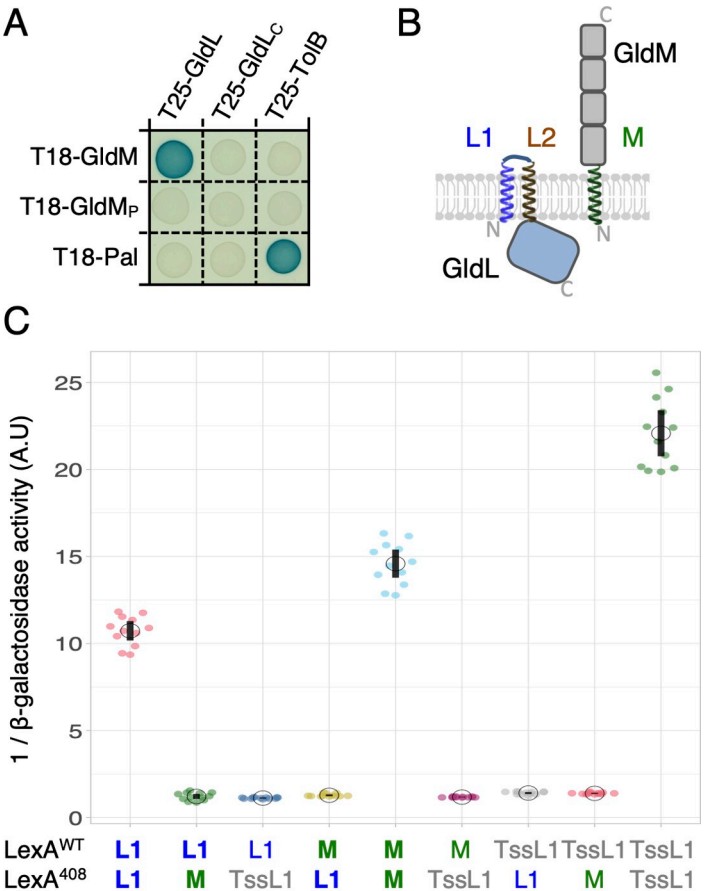

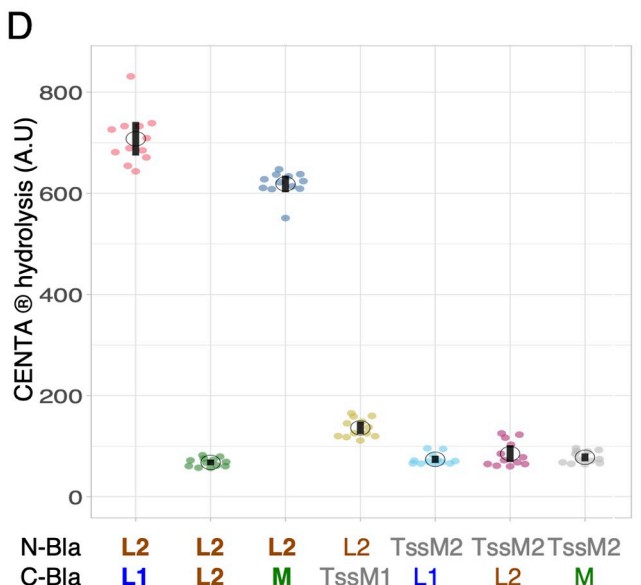

**Fig 2. Interactions between GldL and GldM TMHs. (A)** Bacterial 2-hybrid assay. GldL–GldM interaction is dependent on their TMHs. BTH101 cells producing the indicated proteins (GldL and GldM) or domains (GldL$_C$, cytoplasmic domain of GldL, amino acid 59 to 189; GldM$_P$, periplasmic domain of GldM, amino acid 36 to 513) fused to the T18 and T25 domain of the *Bordetella* adenylate cyclase were spotted on X-Gal-IPTG reporter LB agar plates. The blue color of the colony reports interaction between the 2 partners. Controls include T18 and T25 fusions to TolB

and Pal, 2 proteins that interact but unrelated to the T9SS. **(B)** Schematic representation of GldL and GldM domains and topologies in the IM. The N-terminus (N) and carboxyl terminus (C) of proteins are indicated in gray. The GldL TMH1 (L1, blue) and TMH2 (L2, brown) and GldM TMH (M, green) are indicated. **(C)** Homo- and heterodimerization of in-to-out TMHs of GldL and GldM probed with the GALLEX method. Jitter plots of β-galactosidase activity reporting the dimerization of TMHs fused to LexA[WT] or LexA[408]. Measurements (transcriptional repression) are reported as 1/β-galactosidase activity. Data are combined from technical triplicates of 4 independent measurements (2 colonies from 2 independent transformations each). Interactions with TssL1 (in gray) served as negative control or positive control for self-interactions. TssL1 is the TMH of TssL, a protein of the *E. coli* T6SS that homodimerizes. Data underlying this figure can be found in S1 Data. **(D)** Homo and heterodimerization of the in-to-out and out-to-in THMs of GldL and GldM probed with the BLA method. Jitter plots of CENTA chromogenic substrate hydrolysis after 10 minutes of incubation. The activity is reported as the $A_{405nm}$ value per $A_{600nm}$. Controls include interaction assays with the TMHs of TssM (TssM1 and TssM2, in gray), a subunit of the *E. coli* T6SS. Data underlying this figure can be found in S1 Data. T6SS, type VI secretion system; T9SS, type IX secretion system; TMHs, transmembrane helices; WT, wild-type.

characterization of GldM and its homolog PorM from *P. gingivalis* revealed a conformational flexibility in the periplasmic region of GldM [25]. The extracellular region of GldM assembles as homodimer, spanning most of the periplasm, composed of 4 domains, D1 to D4 [25,58]. Interestingly, the homolog PorM presents a kink between domains D2 and D3 [25], while a kink between the GldM D1 and D2 domains has been proposed in a composite model [42], suggesting that GldM/PorM may alternate between several conformational states. Indeed, recent proteolytic susceptibility assays showed that the PMF regulates conformational changes in GldM: upon PMF dissipation by CCCP, 2 cleavages at the interface of domains D2 and D3 were identified by mass spectrometry after limited trypsinolysis [59]. Here, we extend these observations by showing that the in vivo conformation of GldM is altered by drugs that perturb the proton gradient such as CCCP and nigericin, but remained unaffected upon treatment with the $F_1F_0$ ATPase inhibitors sodium azide and sodium arsenate, nor upon dissipation of the $\Delta\psi$ by valinomycin (Fig 3A), demonstrating that GldM undergoes a structural transition dependent on the IM proton gradient (Fig 3B).

**A conserved glutamate residue in GldL TMH2 is critical for harvesting the proton gradient.** The activity of bacterial MotAB-like molecular motors characterized so far depends on conserved acidic residues located in their TMHs [39,40,48,60]. We first tested the effect of the

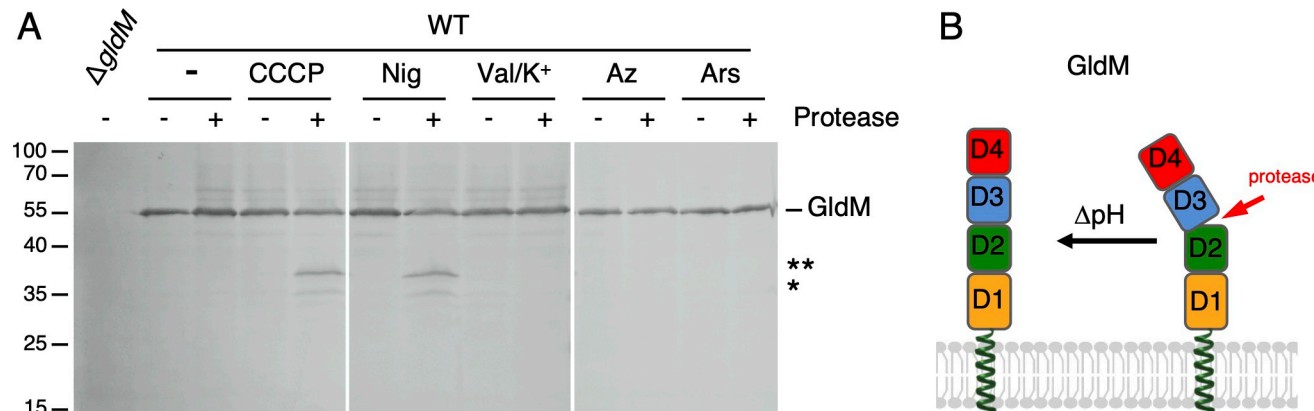

**Fig 3. Conformational changes in GldM periplasmic domain in response to the proton gradient. (A)** GldM protease accessibility assay. Spheroplasts of WT *F. johnsoniae* or the Δ*gldM* mutant were treated (+) or not (−) with the trypsin protease and 10 μM CCCP (PMF inhibitor), 7 μM nigericin (ΔpH inhibitor), 40 μM valinomycine/K+ (Δφ inhibitor), 1.5 mM sodium azide, or 20 mM sodium arsenate ($F_1F_0$ ATPase inhibitors). GldM was analyzed by SDS-PAGE and immunoblot with anti-GldM antibodies. The full-length GldM protein is indicated, as well as degradation products (* and **). The molecular mass markers (in kDa) are indicated on the left. **(B)** Schematic model of GldM conformational transition dependent on the proton gradient. WT, wild-type.

drug N-N'-Dicyclohexyl-carbodimide (DCCD), which covalently reacts with carboxylic groups located in a hydrophobic environment [61]. Addition of DCCD abrogated gliding motility (Fig 4A and 4B) and had the same effect on GldM conformation as CCCP (Fig 4C). This effect was irreversible since DCCD remains covalently bound (Fig 4B). These results therefore suggested that aspartate or glutamate residues are involved in coupling PMF to GldM conformational change and gliding motility. Sequence alignment showed that 3 acidic residues are conserved in the GldL and GldM N-terminal regions (S2A and S2B Fig): a glutamate at position 31 in the vicinity of GldM TMH (GldM-E31) and 2 glutamates in GldL, one located in TMH2 (E49; strictly conserved in all GldL homologs in the OrthoDB database), and one located between TMH2 and the cytoplasmic domain (E59) (S2C Fig). Substitution of GldM E31 (GldM$^{E31A}$) did not exhibit any defect in gliding motility compared to wild-type (WT) GldM, indicating that this residue does not play a significant role in T9SS-dependent secretion or gliding (Fig 4D). By contrast, cells producing GldL$^{E49A}$ failed to adhere to the glass surface, while substitution of GldL E59 abolished gliding motility without affecting adherence (i.e., SprB secretion by the T9SS) (Fig 4E). Interestingly, the GldL$^{E49A}$ and GldL$^{E59A}$ variants presented distinct phenotypes regarding SprB secretion and dynamics as shown by live cell immunolabeling using polyclonal anti-SprB antibodies and fluorescence time-lapse microscopy on agarose pads. As previously reported [6], WT cells exhibited surface-exposed SprB fluorescent foci that describe an overall helicoidal pattern along the long cell axis, with dispersed velocity in the order of 1 μm.s$^{-1}$ (Fig 4F). No fluorescent focus was observed in Δ*gldL* mutant cells, which are unable to secrete SprB, indicating that SprB immunolabeling was specific. A similar observation was made with *gldL*$^{E49A}$ mutant cells (Fig 4F), demonstrating that T9SS-dependent SprB secretion to the cell surface requires residue E49 in GldL TMH2. By contrast, the GldL$^{E59A}$ substitution supported SprB secretion but abolished the dynamic cell surface movements of the adhesin (Fig 4F and 4G). It is noteworthy that all GldL and GldM variants were produced in *F. johnsoniae* at levels comparable to the WT proteins (S2D and S2E Fig), although GldL$^{E49A}$ migrated with lower apparent size than the WT protein (S2E Fig), an aberrant migration already observed in a separate study [42] and likely caused by the difference in detergent binding in SDS-PAGE between the TMH2 variants [62]. Taken together, these results support a model in which GldL E49 is required for secretion of the SprB adhesin and constitutes a key determinant of T9SS, whereas GldL E59 is dispensable for secretion and plays a specific function in gliding because it is only required for SprB movement. We next tested the contribution of these acidic residues for the regulation of GldM conformation. Protease accessibility assays showed that GldL and its Glu49 residue are required to maintain GldM in the conformation required for T9SS activity (Fig 4H). By contrast, the GldL E59A substitution did not impact GldM proteolytic susceptibility (Fig 4H), suggesting that the GldM conformation change observed by limited proteolysis is specifically linked to effector secretion rather than gliding motility.

**Protonation of GldL glutamate residues.** To address the question whether GldL E49 and E59 residues undergo protonation and deprotonation cycles, we determined their pKa values. A $^{15}$N/$^{13}$C Glu-labeled synthetic peptide corresponding to GldL TMH2 (L2, residues Val40 to Val61) was solubilized in deuterated dodecylphosphorylcholine (DPC) micelles and analyzed by nuclear magnetic resonance (NMR). pKa values of 5.54 ± 0.04 and 5.65 ± 0.13 for the carboxylic groups of residues E49 and E59, respectively, were measured by the pH-dependent chemical shifts in two-dimensional $^{13}$C-HSQC experiments (Fig 5A and 5B). In the presence of peptides corresponding to GldL TMH1 (L1, residues Lys6 to Thr29) and GldM TMH (M, residues Leu15 to Leu38), the pKa values slightly increased to 5.83 ± 0.02 for both glutamates (Fig 5B). The behavior of the $^{13}$C chemical shifts for the GldL TMH2 glutamate residues was then monitored in presence of the different peptides. At pH 5.2 (i.e., protonated glutamates),

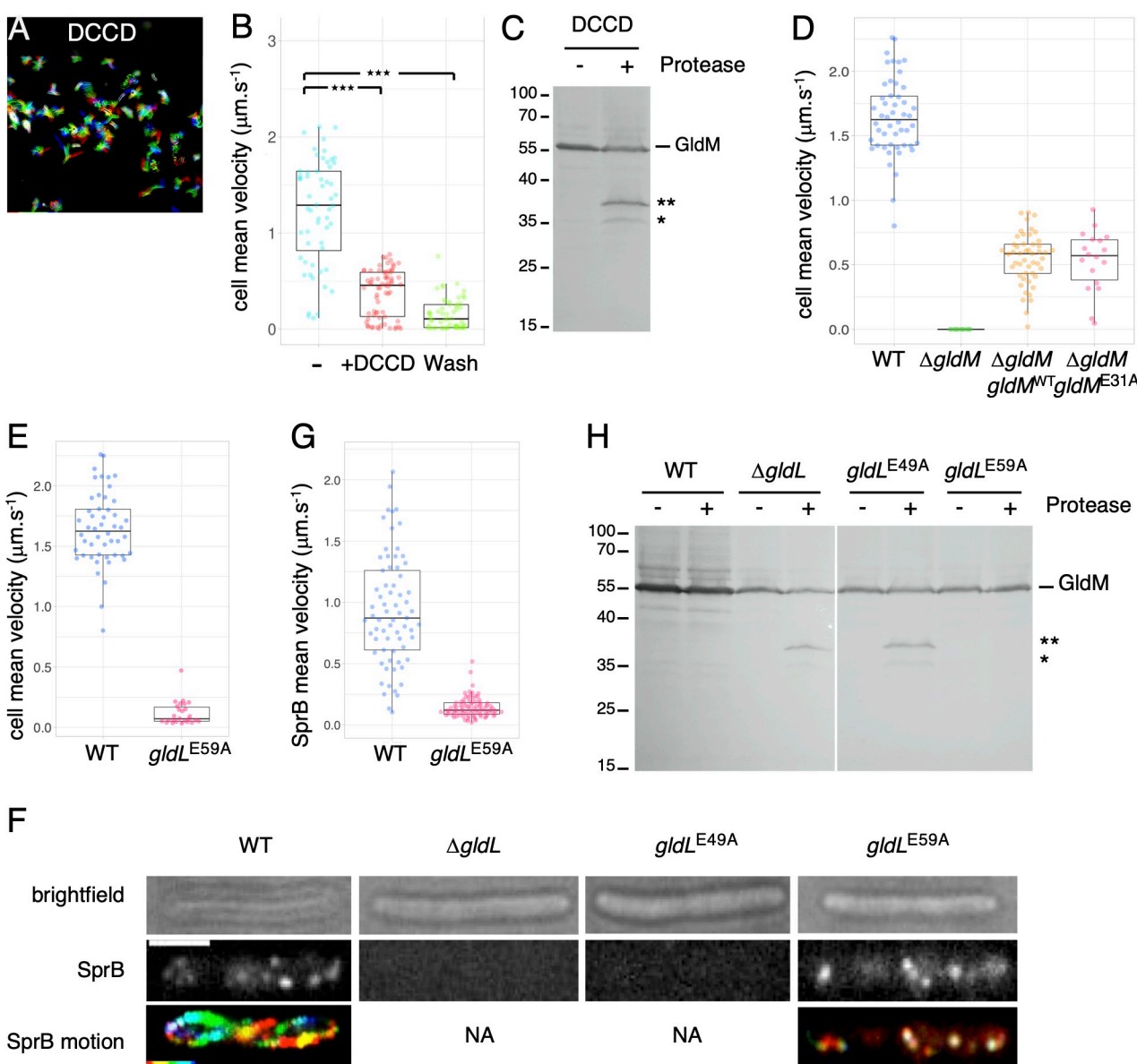

**Fig 4. Roles of conserved GldL TMH2 glutamates in T9SS-dependent secretion and dynamics of SprB. (A)** The addition of the drug DCCD inhibited cell motility. Rainbow traces of cell motility on glass recorded by phase contrast microscopy over time (2 minutes) in the presence of 10 μM DCCD. **(B)** Combined jitter plots/boxplots of mean cell gliding velocity (in μm.s$^{-1}$) of $n > 50$ WT cells before (−), during a pulse of 10 μM DCCD (+DCCD), and after wash with fresh CYE medium (Wash). Statistical significance relative to the nontreated condition (−) is indicated above the plots (ns, nonsignificative, ***, $p < 0.001$; Wilcoxon $t$ test). Data underlying this figure can be found in S1 Data. **(C)** Effect of DCCD on GldM protease susceptibility. GldM protease accessibility assay on spheroplasts of WT *F. johnsoniae* in absence (−) or presence (+) of 10 μM DCCD. **(D)** GldM conserved glutamate 31 does not play an important role in gliding motility. Combined jitter plots/boxplots of mean cell gliding velocity (in μm.s$^{-1}$) of WT cells ($n = 50$), and strains expressing ectopically *gldM*$^{wt}$ ($n = 49$) or *gldM*$^{E31A}$ point mutant ($n = 18$) in a Δ*gldM* mutant background. The Δ*gldM* mutant has been placed in the graph for relevance but cell velocity has not been measured for that strain because it did not adhere to glass. Data underlying this figure can be found in S1 Data. **(E)** Combined jitter plots/boxplots of mean cell gliding velocity (in μm.s$^{-1}$) of WT cells ($n = 50$) and a *gldL*$^{E59A}$ point mutant ($n = 26$). Strains were cultivated in CYE and single-cell gliding was observed on a free glass coverslip by phase contrast microscopy during 2 minutes. Gliding of the Δ*gldL* mutant and of the *gldL*$^{E49A}$ point mutant was not measured because cells did not adhere to the glass substratum. Data underlying this figure can be found in S1 Data. **(F)** Localization and dynamics of SprB on the cell surface in the WT strain, a Δ*gldL* mutant, a *gldL*$^{E49A}$ point mutant, and a *gldL*$^{E59A}$ point mutant. A representative cell is shown. Strains were cultured in CYE and, after SprB immunolabeling, were sandwiched between an agarose pad and a glass coverslip to significantly limit cell movement and facilitate SprB signal acquisition and analysis. SprB was immunolabeled using a primary serum directed against SprB and Alexa-488 fluorescent secondary antibodies. Fluorescence was recorded with 100-ms intervals for several seconds. The brightfield image (top panel), the first frame (middle panel, in gray levels), and the rainbow trace of SprB motion over time (bottom panel, not available for the Δ*gldL* mutant and the *gldL*$^{E49A}$ point mutant) are shown. Scale bar, 2 μm. **(G)** Combined jitter plots/boxplots of mean displacement velocity (in μm.s$^{-1}$) of SprB in WT cells ($n = 69$) and a *gldL*$^{E59A}$ point mutant

($n$ = 85). SprB fluorescent spots were detected and tracked over time (>2 seconds) using the Trackmate plugin. Data underlying this figure can be found in S1 Data. **(H)** GldM protease accessibility assay in WT *F. johnsoniae*, the $\Delta gldL$ mutant, and GldL[E49A] and GldL[E59A] point mutants. Spheroplasts were treated with (+) or not (−) with the trypsin protease. GldM was analyzed by SDS-PAGE and immunoblot with anti-GldM antibodies. The full-length GldM protein is indicated, as well as degradation products (* and **). The molecular mass markers (in kDa) are indicated on the left. DCCD, N-N'-Dicyclohexyl-carbodimide; T9SS, type IX secretion system; WT, wild-type.

we observed chemical shift variations in the presence of L1, M, or both (Fig 5C). These data confirm that GldL TMH2 interacts with GldL-TMH1 and GldM-TMH and that the presence of these TMH peptides influences the environment of the glutamate residues. However, at pH 6.7 (i.e., deprotonated glutamates), no chemical shift was observed upon addition of the L1, M or both peptides (Fig 5C), suggesting that GldL TMH1 and GldM TMH are not in the environment of the glutamate residues. Taken together, these results suggest that the protonation state of the glutamic acids regulates contacts between TMH2 and the other TMHs in the GldLM complex and hence that GldLM helix organization is likely to be modified during motor function, as evidenced for the MotAB and TolQR motors [63,64].

Altogether, our results support a model in which GldL and GldM form an IM proton channel with conserved critical glutamates that are protonated and deprotonated in response to the proton gradient to power both T9SS-dependent secretion and gliding motility. Our results also

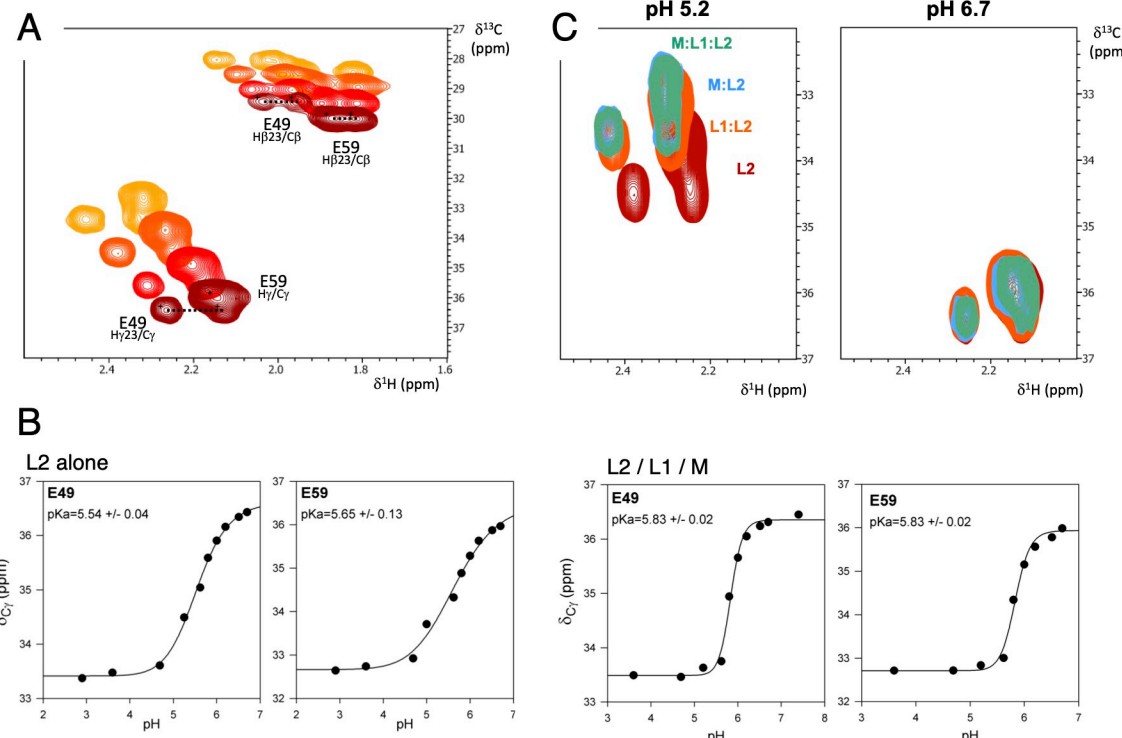

**Fig 5. GldL glutamate residues protonation probed by NMR.** pKa determination of $^{13}$C-Glu of free and complexed L2 peptide. **(A)** Two-dimensional $^{13}$C-HSQC spectra of 1 mM L2 peptide ($^{13}$C-Glu labeled) in 150 mM deuterated DPC in 50 mM phosphate buffers at different pH (pH 2.9 (yellow), 5.0 (orange), 5.8 (red), 6.7 (brown)). **(B)** The pH dependent chemical shift variations of $C_\gamma$ carbons of E49 and E59 of the L2 peptide free or complexed with L1 and M in a 1:1:1 molar ratio, were measured, fitted, and apparent pKa values were calculated using the Henderson–Hasselbach equation. Data underlying this figure can be found in S1 Data. **(C)** Two-dimensional $^{13}$C-HSQC spectra of 1 mM L2 peptide ($^{13}$C-Glu labeled) in 150 mM deuterated DPC in 50 mM phosphate buffer at pH 5.2 (left panel) and pH 6.7 (right panel), in the absence (brown) and presence at molar ratio 1:1 of GldL-TMH1 peptide (L1, orange), GldM-TMH peptide (M, blue), and both L1 and M peptides (green). DPC, dodecylphosphorylcholine; NMR, nuclear magnetic resonance.

demonstrate that the protonation state of GldL E49 controls changes within the GldLM TMHs packing that are likely transmitted to the GldM periplasmic domain.

## GldLM motors and SprB adhesin do not have the same dynamics

SprB adhesins follow a closed right-handed helical track at the cell surface [5,6]. Two models have been proposed on how the T9SS controls SprB motion [3,4]. In the first model, fixed rotary motors may activate treads to which SprB adhesins are connected. In the second model, SprB adhesins are directly connected to moving GldLM motors. The first scenario requires a network of motor complexes along the SprB helicoidal path. The second scenario implies that SprB colocalizes with dynamic GldLM complexes and that SprB and GldLM move concomitantly along the helicoidal path. To explore these possibilities, we characterized the localization of the GldLM motor complex. Structured illumination microscopy (SIM) recordings of *F. johnsoniae* fixed and permeabilized cells immunolabeled with polyclonal primary antibodies against the GldL, GldM, GldK, or GldN proteins, and fluorescent secondary antibodies showed that each protein was distributed in many foci along the cell body (Fig 6A). For example, we numbered 33 ± 10 GldL foci per cell in average (*n* = 11, Fig 6A). These data suggest that multiple Gld motors decorate the cell envelope. A similar number of GldM foci were numbered (27 ± 5), in agreement with the assembly of functional GldLM complexes. For GldK and GldN, 13 ± 7 and 12 ± 4.5 foci were observed in average per cell, respectively, again in agreement with the formation of GldKN ring complexes. Based on the higher number of GldL/M foci as compared to GldK/N foci, one may hypothesize that GldLM complexes may exist in 2 subpopulations, one free and one associated with GldKN rings. To provide further information, we sought to perform live observations. However, none of the plasmid-borne or chromosomal fluorescent fusions to GldL or GldM we generated supported WT gliding, possibly due to the size of the fluorescent protein tags. We therefore turned to a more sophisticated method to generate functional and time-trackable proteins, using the alfa technology [65].

The alfatag is a 13–amino acid peptide that is specifically and almost irreversibly bound by the NBalfa nanobody with an affinity of approximately 0.26 pM [65]. The sequence encoding the alfatag was introduced in frame at the carboxyl terminus of the GldL-coding sequence at the native locus. GldL-alfa was functional and supported single-cell gliding (S3A and S3B Fig). In addition, no cleavage of the alfa tag was observed (S3C Fig). We then engineered a replicative plasmid expressing NBalfa-sfGFP under the control of an IPTG-inducible promoter in *F. johnsoniae*. Expression of NBalfa-sfGFP did not perturb cell gliding, either in a WT background or in a strain expressing GldL-alfa (S3B Fig). NBalfa-sfGFP was diffuse in WT cells that do not express GldL-alfa (S3D Fig). By contrast, NBalfa-sfGFP exhibited a punctate pattern in GldL-alfa cells (Fig 6B). Distinct foci were visible as well as more patchy signals, rendering quantification difficult. Remarkably, these foci were not all static relative to the cell, as opposed to the SprA translocon [26], nor did they behave like SprB adhesins that travel along the entire cell length. Time-lapse microscopy and kymograph analyses of signal dynamics indicated that some foci remain static while others moved quickly but at varying speed relative to the cell (Fig 6B, S1 Movie). In addition, fluorescence recovery after photobleaching (FRAP) experiments in nonmoving cells suggest that GldL-alfa movement was restricted to short distances within the cell because fluorescent signal could not be recovered over a large bleached cell region (Fig 6C and 6D). Based on these results and on the number of GldLM and GldKN foci per cell, we propose that these 2 GldL populations could correspond to GldLM complexes engaged into static complexes with the OM translocon to secrete substrates and to free GldL proteins or to GldLM complexes following a track to energize SprB motion. However, colocalization experiments in live cells with immunolabeled SprB showed that GldL-alfa and SprB do

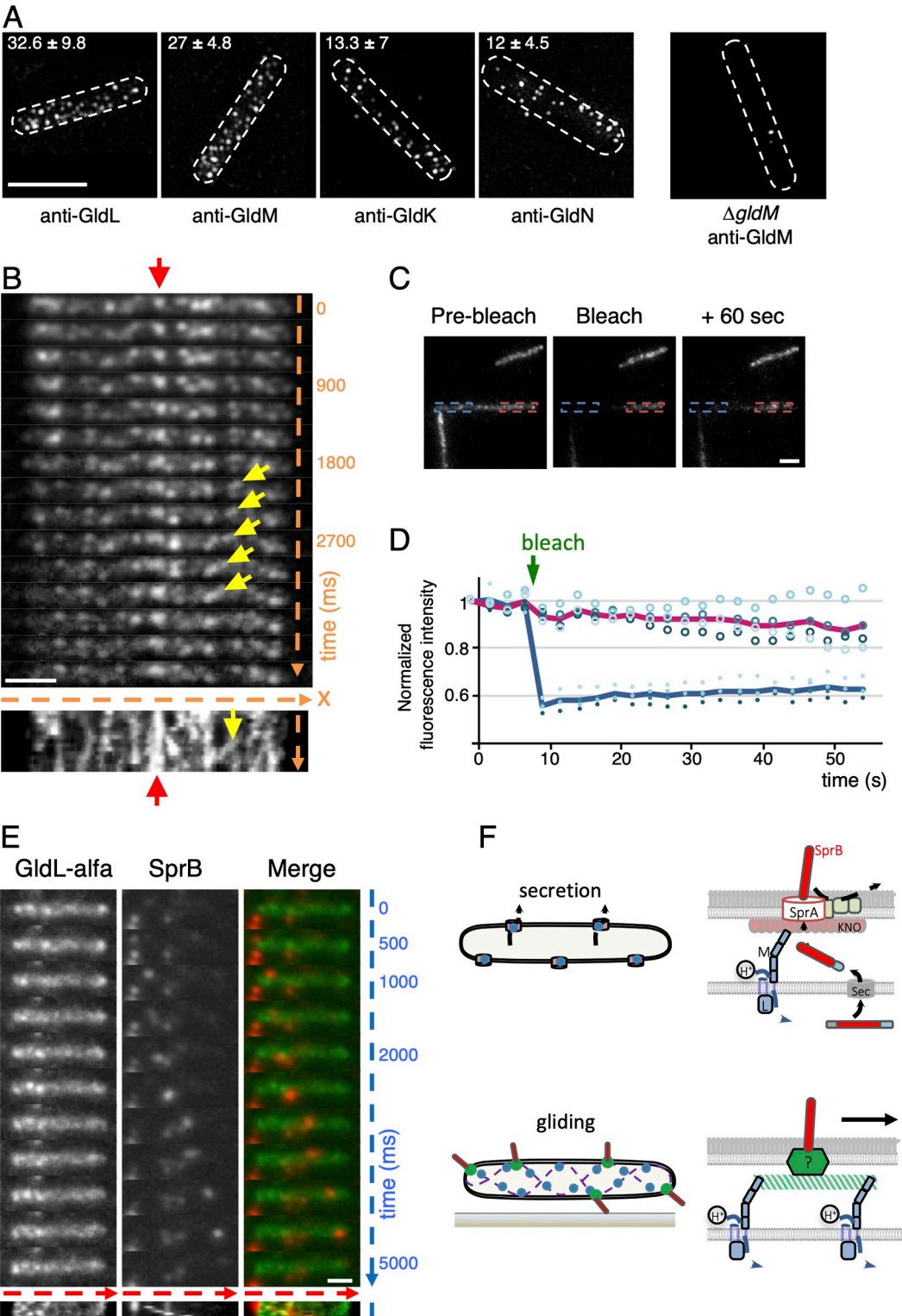

**Fig 6. Dynamic localization of Gld complexes. (A)** Localization of Gld proteins in fixed cells. Immunostaining of GldL, GldM, GldK, and GldN and observation by SIM. A typical specificity control is shown on the right panel with a Δ*gldM* mutant stained with antibodies directed against GldM. Scale bar, 2 μm. The mean number and standard deviation of foci per cell is indicated on top (*n* > 10). **(B)** Live cell dynamics of a functional GldL-alfa fusion bound to NBalfa-sfGFP. NBalfa-sfGFP expression was induced with 1 mM IPTG for 1 hour prior to observation. GldL-alfa dynamics was then followed by

time-lapse fluorescence microscopy with 300-ms intervals. For a representative cell, a stack of individual frames is shown. Time is indicated in ms. At the bottom of the stack, a kymograph of the fluorescence signal in the same representative cell. The x-axis is the position of GldL-alfa/NBalfa-sfGFP signal with respect to the substratum (glass), and the y-axis is time. The red arrow shows an example of a static signal, and the yellow arrow points to an example of a moving GldL-alfa focus. Scale bar, 2 μm. **(C)** GldL-alfa does not travel long distance within the cell. Diffusion of GldL-alfa/NBalfa-sfGFP fluorescence signal over time assayed by FRAP. Cells expressing GldL-alfa and NBalfa-sfGFP were pulse-bleached in the region indicated by the blue rectangle. A representative cell is shown. Scale bar, 2 μm. **(D)** Fluorescence intensities were measured in the bleached region (blue rectangle) and a nonbleached region (red rectangle) for 60 seconds in 4 cells. Individual measurements are shown with spots (filled spots for bleached regions and empty spots for nonbleached regions). The green arrow indicates the moment of bleaching. The fluorescence intensity of each region of interest was normalized to the first prebleached intensity. The blue (bleached region) and red (nonbleached) lines indicate the average of all measurements. Data underlying this figure can be found in S1 Data. **(E)** GldL-alfa and SprB do not colocalize. NBalfa-sfGFP expression was induced with 1 mM IPTG for 1 hour prior to observation. Then, dynamic localization of GldL-alfa bound to NBalfa-sfGFP (in green) and immunostained SprB (in red) in a representative cell. Fluorescence was followed by hilo microscopy with 500-ms intervals. A stack of individual frames is shown. Time is indicated in ms. At the bottom of the stack, a kymograph of the fluorescence signal in the same representative cell. The x-axis is the position of the cell with respect to the substratum (glass), and the y-axis is time. Scale bar, 1 μm. **(F)** Model of GldLM molecular motors function in type IX secretion (top) and surface adhesin dynamics (bottom). GldLM motors (blue) are fueled by the proton gradient (H+), leading to conformational shifts of the periplasmic domain of GldM. When associated with the T9SS (top), GldLM motors generate mechanical torque to rotate a GldKNO ring to drive secretion of SprB (red) through the SprA translocon. GldLM motors may also be associated with the gliding machinery (bottom), in which they serve to displace SprB on the cell surface via the displacement of a track or baseplate machinery (in green). FRAP, fluorescence recovery after photobleaching; SIM, structured illumination microscopy; T9SS, type IX secretion system.

not follow the same trajectories (Fig 6E). These results support the tread model [66,67] in which GldLM proton channels convert the proton gradient into mechanical force to displace or activate treads involved in SprB movement (Fig 6F).

## Concluding remarks

In this study, we provided evidence that type IX secretion and surface adhesin motion are energized by a molecular motor fueled by the proton gradient, like the flagellar motor and other bacterial molecular motors. Our data support the idea that interactions between the TMHs of GldL and GldM shift in response to the proton flux, eventually leading to conformation changes in the GldM periplasmic domain. Conserved glutamate residues in GldL are important in this process but are not equivalent in terms of function. While amino acid E49 in GldL is essential for secretion through the T9SS as observed by Hennell and colleagues [42], glutamate at position 59 is only required for gliding, indicating that in *F. johnsoniae*, T9SS secretion itself does not require concomitant SprB motion along the cell surface. Thus, it is tempting to speculate that secretion and SprB motion are not supported by the same mechanical rearrangements in GldM or that SprB motion may require more mechanical torque than the secretion process. Furthermore, since SprB motion and secretion are uncoupled in the GldL E59A point mutant, this mutation is an interesting tool to study secretion independently of gliding in *F. johnsoniae*. Nevertheless, GldL-E59 is also conserved in nongliding bacteria like *P. gingivalis*, suggesting that it may also serve for T9 secretion in other bacteria (S2A Fig).

Our data also support the idea that GldM conformational shift upon PMF sensing could be converted into mechanical torque through the periplasmic part of the T9SS. Indeed, we showed that GldM periplasmic domain is connected to the T9SS GldKNO subcomplex, similar to the PorKLMN complex in *P. gingivalis*. Two recent studies help understand how this could work. First, the structure of the GldLM motor showed that 10 GldL TMHs (5 GldL molecules) wrap 2 GldM TMHs in an asymmetric manner [42]. By similarity to other MotAB-like motors, it was proposed that GldM TMHs would rotate within a GldL ring in response to the PMF to generate mechanical movement of GldM periplasmic domain. These findings are consistent with our data and provide an explanation for why GldL-E49 is required for motor function.

However, they do not explain the role of E59, which is located outside the membrane in the GldLM structure (S2C Fig) [42]. One may hypothesize that E59 enters the proton channel when GldM rotates. Second, in situ PorKN rings were observed by cryo-electron tomography [59,68]. These rings may serve to maintain T9SS subcomplexes in close proximity to allow sequential translocation, maturation, and attachment of the substrates [68]. Therefore, an attractive hypothesis is that GldM conformational changes in response to the proton gradient could generate mechanical torque for the rotation of GldKNO rings, similar to cogwheels, which directly or indirectly facilitate secretion of T9SS substrates.

Finally, our results are consistent with the "rack and pinion" model proposed by Shrivastava and Berg to explain how the GldLM complex participates in SprB displacement [66]. Our microscopy data suggest the existence of static GldLM motors, which are presumably associated with static T9SS translocons, and GldLM complexes that are dynamic but that move differently than do SprB molecules. These motors could be linked to unidentified motion treads carrying SprB adhesins.

## Material and methods

### Bacterial strains, media, and chemicals

All strains are listed in S1 Table. *Escherichia coli* strains DH5α and BTH101 were used for cloning procedures and bacterial 2-hybrid assay, respectively. *E. coli* cells were grown in Lysogeny Broth, at 37°C or 28°C. For BACTH experiments, gene expression was induced by the addition of iso-propyl-β-D-thio-galactopyranoside (IPTG, Sigma-Aldrich, St. Louis, MI, USA, 0.5 mM), and plates were supplemented with 5-bromo-4-chloro-3-indolyl-β-D-galactopyranoside (X-Gal, Eurobio, Les Ullis, France, 40 μg.mL$^{-1}$). *F. johnsoniae* CJ1827, a streptomycin-resistant *rpsL*2 derivative of ATCC 17061 (UW101), was used as model micro-organism. *F. johnsoniae* cells were grown at 28°C in Casitone Yeast Extract (CYE) medium [69] or Motility Medium (MM) [70] as indicated. For selection and maintenance of the antibiotic resistance, antibiotics were added to the media at the following concentrations: erythromycin, 100 μg.mL$^{-1}$; streptomycin, 100 μg. mL$^{-1}$; tetracycline, 20 μg.mL$^{-1}$, ampicillin, 100 μg.mL$^{-1}$, kanamycin, 50 μg.mL$^{-1}$, and chloramphenicol, 40 μg.mL$^{-1}$. Specific enzyme and chemicals source were as follows: trypsin (Sigma-Aldrich), CCCP (Sigma-Aldrich, 10 μM), nigericin (Nig, Sigma-Aldrich, 7 μM), valinomycine (Val, Sigma-Aldrich, 40 μM), sodium azide (Az, Sigma-Aldrich, 1.5 mM), arsenate (Ars, Sigma-Aldrich, 20 mM), and DCCD (Sigma-Aldrich, 100 μM).

### Genetic constructs

All plasmids and oligonucleotide primers used in this study are listed in S1 Table. Enzymes for PCR and cloning were used as suggested by manufacturers.

Chromosomal mutants were generated as described [71]. The suicide plasmid designed to generate an in-frame deletion of *gldM* was built as follows. A 2.5-kb fragment containing the region upstream of *gldM* and *gldM* start codon was PCR amplified using oligonucleotide primers F1-ΔgldM and R1-ΔgldM. This fragment was digested with *Bam*HI and *Sal*I and inserted into plasmid pRR51 cut with the same restriction enzymes to generate an intermediate plasmid. A 2.5-kb fragment containing *gldM* stop codon and the region downstream of *gldM* was PCR amplified using oligonucleotide primers F2-ΔgldM and R2-ΔgldM. Similarly, it was digested with *Sal*I and *Sph*I and inserted into the previously generated plasmid cut with *Sal*I and *Sph*I to generate plasmid pRR51-Δ*gldM*.

The suicide plasmids designed to build *gldL*$^{E49A}$ and *gldL*$^{E59A}$ strains were constructed as follows. A plasmid with plasmid pRR51 backbone and carrying a 4-kb region centered around *gldL* E49A and E59A codon substitutions was synthesized (Geneart, Thermo Fisher Scientific, Waltham, MA, USA). The suicide plasmid pRR51-*gldL*$^{E49A}$ was then built by restoring the

codon for E59 in the previously synthesized plasmid by site directed mutagenesis with oligonucleotide primers Fw-GldL-A59E and Rv-GldL-A59E. Similarly, the suicide plasmid pRR51-$gldL^{E59A}$ was built by restoring the codon for E49 in the previously synthesized plasmid by site directed mutagenesis with oligonucleotide primers Fw-GldL-A49E and Rv-GldL-A49E.

The suicide plasmid used to generate the GldL-alfa fusion was made as follows. A 1.5-kb fragment containing *gldL* and part of the alfa tag was PCR amplified using oligonucleotide primers oTM582 and oTM590. A 1.5-kb fragment containing part of the alfa tag, a stop codon, and the region immediately downstream of *gldL* stop codon was PCR amplified using oligonucleotide primers oTM591 and oTM592. These fragments were assembled using Gibson isothermal reaction and reamplified using oligonucleotide primers oTM582 and oTM592. This fragment was digested with *Bam*HI and *Sph*I and inserted into pRR51 cut with the same restriction enzymes. Plasmids for GALLEX and BLA were engineered by hybridizing complementary oligonucleotides corresponding to the GldL or GldM TMHs and inserting them into *Nhe*I-*Bam*HI-digested target GALLEX or BLA vectors. BACTH plasmids were engineered by restriction and ligation as previously described [23]. The replicative plasmid designed for complementation and production of GldLM$^{WT}$ were constructed as follows. A fragment containing *gldL* and *gldM* open reading frames was PCR amplified using oligonucleotide primers 5-BamHI-LM and 3-XbaI-LM. This fragment was digested with *Bam*HI and *Xba*I and inserted into plasmid pCP11 [72] cut with the same restriction enzymes. The replicative plasmid designed to produce GldM$^{E31A}$ was then generated by quick change site directed mutagenesis using oligonucleotide primers 5-GldM-E31A and 3-GLdM-E31A.

The replicative plasmid designed to express NbAlfa-sfGFP from an IPTG inducible promoter in *F. johnsoniae* was designed as follows. A first replicative plasmid was built with an IPTG-inducible promoter, a multicloning site and *lacI* constitutive expression for repression in the absence of IPTG in *F. johnsoniae*. A fragment containing the promoter of Fjoh_0697 [73] with *lacO3* and *lacO1* operator sites flanking the −33 and −7 promoter sequences, pCP23 multicloning site and Fjoh_0139 promoter after the *Pst*I restriction site was synthesized (Geneart, Thermo Fisher Scientific). A fragment containing *lacI* open reading frame was PCR amplified using oligonucleotide primers oTM495 and oTM496. These fragments were assembled by Gibson isothermal reaction and reamplified using oligonucleotide primers oTM497 and oTM496. This fragment was digested with *Kpn*I and *Sph*I and inserted into pCP23 [69] cut with the same restriction enzymes to generate plasmid pCP-*lac*. Then, the gene encoding NbAlfa was synthesized (Geneart, Thermo Fisher Scientific) and reamplified using oligonucleotide primers oTM612 and oTM596. sfGFP, codon-optimized for translation in *F. johnsoniae*, was also synthesized (Geneart, Thermo Fisher Scientific) and then reamplified by PCR using oligonucleotide primers oTM595 and oTM593. These fragments were assembled using the Gibson isothermal reaction and reamplified using oligonucleotide primers oTM612 and oTM593. It was then digested with *Bam*HI and *Nhe*I and inserted into plasmid pCP-*lac* cut with the same restriction enzymes.

## Protein interaction assays

The adenylate cyclase-based bacterial 2-hybrid technique was used as previously published [23]. Briefly, the proteins to be tested were fused to the isolated T18 and T25 catalytic domains of the *Bordetella* adenylate cyclase. After introduction of the 2 plasmids producing the fusion proteins into the BTH101 reporter strain, plates were incubated at 28°C for 24 hours. Three independent colonies for each transformation were inoculated into 600 μL of LB medium supplemented with ampicillin, kanamycin, and IPTG (0.5 mM). After overnight growth at 28°C, 10 μL of each culture was spotted onto LB plates supplemented with ampicillin, kanamycin,

IPTG, and X-Gal and incubated at 28˚C. Controls include interaction assays with TolB and Pal, 2 protein partners unrelated to the T9SS. The experiments were done in triplicate and a representative result is shown.

GALLEX and BLA were performed as described [55].

### Protease susceptibility assay

*F. johnsoniae* cells were grown in 5 mL of CYE medium to an $A_{600}$ = 0.8, harvested by centrifugation and resuspended in 100 μL of 20 mM Tris-HCl pH 8.0, 20% sucrose, 1 mM EDTA, and 100 μg.mL$^{-1}$ of lysozyme. After 30-minute incubation at room temperature (20˚C), 100 μL of ice-cold sterile water was added and the mixture was carefully mixed by 3 inversions. Moreover, 50 μL of each spheroplast suspension was treated with trypsin (100 μg.mL$^{-1}$). After 5 minutes on ice, 17 μL of boiling 4× Laemmli loading buffer was added and immediately vortexed and boiled for 5 minutes prior to SDS-PAGE and immunoblot.

### Western blot analyses

*F. johnsoniae* cells were grown to mid-log phase in CYE at 28˚C. Whole cells were prepared for SDS-PAGE and western blotting assays were performed as previously described. Equal amounts of total proteins were loaded for each sample based on culture optical densities. Anti-GldL, anti-GldM [8], and anti-FLAG (Sigma-Aldrich, clone M2) antisera were used at 1/5,000, 1/5,000, and 1/10,000 dilutions, respectively.

### Nuclear magnetic resonance

NMR experiments were carried out on a Bruker Avance III 600 MHz spectrometer, at 300 K. Three synthetic peptides L1 (GldL-TMH1: KKVMNFAYGMGAAVVIVGALFKITKK), L2 (GldL-TMH2: KKVMLSIGLLT**E**ALIFALSAF**E**PVKK), and M (GldM-TMH: KKLMYLV-FIAMLAMNVSK**E**VISAFGLKK), with $^{15}$N/$^{13}$C-Glu-labeled, have been studied free and in complexes at molar ratio 1:1 or 1:1:1. NMR samples containing 1 mM peptide concentration in 150 mM deuterated DPC were used in different phosphate buffers (50 mM). The behavior of the $^{13}$C chemical shifts for glutamate residues in the different peptides as a function of pH (2.9 to 8.9) was monitored using a two-dimensional $^{13}$C-HSQC experiment. Chemical shift values as a function of pH were analyzed according to a single titration curve of the form

$$\delta = \delta_{HA} - \left( \frac{(\delta_{HA} - \delta_A)}{1 + 10^{n(pKa-pH)}} \right),$$

where $\delta$ is the observed chemical shift at a given pH, $\delta_{HA}$ and $\delta_A$ are the chemical shifts for the various protonated forms of the peptide, and $n$ is the number of protons transferred.

### Fluorescence microscopy and image analysis

**General microscopy.** For single-cell gliding on glass, cells were grown in CYE at 28˚C to an $A_{600\,nm} \approx 0.7$. Cells were diluted to an $A_{600\,nm} \approx 0.05$ and 100 μL were spotted into μ-Slide chambers with glass coverslip bottom (Ibidi). After 5-minute incubation, floating cells were washed out with fresh CYE medium and gliding of adherent cells was monitored by phase contrast microscopy on a Nikon Eclipse TE-2000 microscope equipped with a 100× NA 1.3 Ph3 objective, a perfect focus system to maintain the plane in focus, and an Orcaflash 4.0 LT digital camera (Hamamatsu Photonics, Shizuoka, Japan). GldL-alfa/NBalfa-sfGFP localization was observed by Hilo microscopy. Cells were grown in CYE overnight without shaking at 28˚C. NBalfa-sfGFP expression was induced with 1 mM IPTG for 1 hour prior to observation. Cells

were spotted on a 2% low-melting agarose pad for immediate observation. Hilo fluorescence microscopy and FRAP experiments were performed with a Nikon Eclipse Ti2 microscope equipped with a 100x NA 1.45 Ph3 objective, an Orca-Fusion digital camera (Hamamatsu Photonics), a perfect focus system, and an Ilas2 TIRF/FRAP module (Gataca Systems, Massy, France).

**Immunolabeling and SIM acquisition.** SprB immunolabeling on live cells was performed essentially as described [6]. Briefly, 500 μL of cells were incubated 5 minutes with a 1/100 dilution of antiserum directed against SprB [1]. Cells were washed once with CYE and further incubated 5 minutes with Alexa488- or Alexa561-labeled anti-rabbit secondary antibodies (Thermo Fisher Scientific). Cells were washed 4 times in CYE and concentrated 5-fold. In order to facilitate SprB detection and tracking during short periods, cells were spotted on a 2% low-melting agarose pad for immediate observation. Immunolabeling of GldL, GldM, GldK, and GldN was performed on fixed cells as previously described [74], except cells were manipulated in tubes instead of on glass slides. Polyclonal antisera directed against GldL, GldM, GldK, or GldN [8] were used at 1/2,000 dilution and further recognized by Alexa488-labeled anti-rabbit secondary antibodies (Thermo Fisher Scientific). SIM was performed on a DeltaVision OMX SR microscope (GE Healthcare, Chicago, IL, USA). The experiments were done in triplicate, and a representative result is shown.

**Image analysis.** Images were analyzed using ImageJ (http://imagej.nih.gov/ij). The MicrobeJ plugin [75] was used to detect and track cells during gliding. The Trackmate plugin [76] was used to detect SprB fluorescence and analyze its dynamics. Statistical dataset analysis was performed using Excel and the R software environment (https://www.r-project.org). Kymographs were generated using the KymoResliceWide plugin (https://imagej.net/KymoResliceWide; E. Katrukha and L. Young). For fluorescence recovery quantification, images were corrected for bleaching using histogram matching prior to signal recovery quantification.

## Supporting information

**S1 Fig. Network of interactions between proteins of the T9SS core components.** Bacterial 2-hybrid assays. **(A)** T9SS OM-associated core complex (GldK, GldN and GldO) and GldJ. **(B)** T9SS OM-associated core complex (GldK, GldN, and GldO), GldJ and the IM-associated core complex (GldM and GldL). The signal sequence was omitted in the constructs for GldN and GldO. The signal sequence and the acylated N-terminal cysteine residue of the mature form were omitted for GldK and GldJ. BTH101 reporter cells producing the indicated proteins or domains (GldL$_C$, cytoplasmic domain of the GldL protein; GldM$_P$, periplasmic domain of the GldM protein) fused to the T18 or T25 domain of the *Bordetella* adenylate cyclase were spotted on plates supplemented with IPTG and the chromogenic substrate X-Gal. The TolB-Pal interaction serves as positive control. **(C)** Model of the interactions between T9SS components defined by bacterial 2-hybrid assay. IM, inner membrane; IPTG, iso-propyl-β-D-thio-galacto-pyranoside; OM, outer membrane; T9SS, type IX secretion system.
(TIFF)

**S2 Fig. (A)** Sequence alignments of the N-terminal regions that encompass the 2 transmembrane segments of GldL homologs. The alignment was performed using TCOFFEE. Red arrows indicate the conserved acidic residues. **(B)** Sequence alignments of the region that encompasses the single transmembrane segment of GldM homologs. The alignment was performed using TCOFFEE. The red arrow indicates the conserved acidic residue. The TMH regions (as defined by James and colleagues [42]) are indicated above the alignments. **(C)** Highlight of GldL-E49 (orange) and E59 (pink) glutamate residues in the structural model of

the GldLM complex. The left panel shows a side view and the right panel shows a view from the cytoplasm. GldL TMHs are colored green. GldM subunits (TMH and first periplasmic domain) are colored blue. **(D)** Western blot analysis of GldM production using anti-GldM antibodies in a Δ*gldM* mutant, WT *F. johnsoniae*, GldM WT or GldM E31A expressed from a plasmid in a Δ*gldM* mutant background. **(E)** Western blot analysis of GldL production using anti-GldL antibodies in the Δ*gldL* mutant (Δ*gldL*), WT *F. johnsoniae*, and strains expressing GldL$^{WT}$-flag (GldL$^{WT}$) or GldL$^{E49A}$-flag (E49A) or GldL$^{E59A}$-flag (E59A). Extracts of cells were subjected to SDS-PAGE and immunodetection with anti-GldL and anti-Flag primary antibodies and HRP-coupled secondary antibodies. Molecular mass markers (in kDa) are indicated on left. TMHs, transmembrane helices; WT, wild-type.
(TIFF)

**S3 Fig. GldL-alfa supports cell gliding. (A)** Rainbow traces of cell motility on glass recorded by phase contrast microscopy over time (2 minutes) in a WT strain and a strain expressing *gldL-alfa* at the native locus. Individual frames from time-lapse acquisition were colored from red (start) to yellow, green, cyan, and blue (end) and merged into a single rainbow image. Scale bar, 20 μm. **(B)** Combined jitter plots/boxplots of mean cell gliding velocity (in μm.s$^{-1}$) of *gldL-alfa* cells in the absence of NBalfa-sfGFP (−, *n* = 124) or with 1 mM IPTG induction of NBalfa-sfGFP for 1 hour (+, *n* = 125) or WT cells in the absence of NBalfa-sfGFP (−, *n* = 135) or with 1 mM IPTG induction of NBalfa-sfGFP for 1 hour (+, *n* = 151). Data underlying this figure can be found in S1 Data. **(C)** Western blot analysis of GldL-alfa production using anti-GldL polyclonal antibodies or anti-alfa NBalfa nanobodies in WT *F. johnsoniae*, in the Δ*gldL* mutant (Δ*gldL*) or in the strain expressing *gldL-alfa* at the native locus. Molecular mass markers (in kDa) are indicated on right. **(D)** Representative micrograph of cells expressing fluorescent NBalfa-sfGFP in a WT background. Scale bar, 2 μm. IPTG, iso-propyl-β-D-thiogalactopyranoside; WT, wild-type.
(TIFF)

**S1 Table. Strains, plasmids and oligonucleotides used in this study.**
(DOC)

**S1 Data. Numerical data underlying relevant figures.** Excel file containing sheets for each corresponding figure.
(XLSX)

**S1 Movie. Live cell dynamics of a functional GldL-alfa fusion bound to NBalfa-sfGFP.** This movie corresponds to the time course shown in Fig 6B. NBalfa-sfGFP expression was induced with 1 mM IPTG for 1 hour prior to observation. GldL-alfa dynamics was then followed by time-lapse fluorescence microscopy with 300-ms intervals.
(AVI)

## Acknowledgments

We thank the members of the Cascales team for insightful discussions and support; Jean-Pierre Duneau for discussion regarding peptide solubilization; Moly Ba, Isabelle Bringer, Annick Brun, Olivier Uderso, Mathilde Valade, and Audrey Gozzi for technical assistance; and Sepah Bonbercq for encouragement.

## Author Contributions

**Conceptualization:** Maxence S. Vincent, Eric Cascales, Thierry Doan.

**Formal analysis:** Maxence S. Vincent, Corinne Sebban-Kreuzer, Thierry Doan.

**Funding acquisition:** Eric Cascales.

**Investigation:** Maxence S. Vincent, Caterina Comas Hervada, Corinne Sebban-Kreuzer, Hugo Le Guenno, Maïalène Chabalier, Eric Cascales, Thierry Doan.

**Methodology:** Maxence S. Vincent, Caterina Comas Hervada, Thierry Doan.

**Project administration:** Maxence S. Vincent, Eric Cascales, Thierry Doan.

**Resources:** Maxence S. Vincent, Artemis Kosta, Françoise Guerlesquin, Tâm Mignot, Mark J. McBride, Eric Cascales, Thierry Doan.

**Supervision:** Eric Cascales, Thierry Doan.

**Validation:** Maxence S. Vincent, Corinne Sebban-Kreuzer, Eric Cascales, Thierry Doan.

**Visualization:** Maxence S. Vincent, Corinne Sebban-Kreuzer, Hugo Le Guenno, Eric Cascales, Thierry Doan.

**Writing – original draft:** Maxence S. Vincent, Eric Cascales, Thierry Doan.

**Writing – review & editing:** Maxence S. Vincent, Corinne Sebban-Kreuzer, Tâm Mignot, Mark J. McBride, Eric Cascales, Thierry Doan.

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
