## [Editor Report · Decision Letter 0]

15 Oct 2021

Dear Dr. Doan, 

Thank you for submitting your manuscript entitled "Dynamic proton-dependent motors power Type IX secretion and gliding adhesin movement in Flavobacterium" for consideration as a Research Article by PLOS Biology.

Your manuscript has now been evaluated by the PLOS Biology editorial staff, as well as by an academic editor with relevant expertise, and I am writing to let you know that we would like to send your submission out for external peer review.

Please note, however, that the outcome of our discussion of your manuscript is that given that the Nature Microbiology paper was published more than 6 months ago, we would need to be persuaded by the reviewers that the paper is technically sound in order to pursue it further for PLOS Biology. This means that we would not be prepared to welcome a revision of the paper if the reviewers considered that extensive revisions were needed. 

Once your full submission is complete, your paper will undergo a series of checks in preparation for peer review. Once your manuscript has passed the checks it will be sent out for review. 

If your manuscript has been previously reviewed at another journal, PLOS Biology is willing to work with those reviews in order to avoid re-starting the process. Submission of the previous reviews is entirely optional and our ability to use them effectively will depend on the willingness of the previous journal to confirm the content of the reports and share the reviewer identities. Please note that we reserve the right to invite additional reviewers if we consider that additional/independent reviewers are needed, although we aim to avoid this as far as possible. In our experience, working with previous reviews does save time. 

If you would like to send your previous reviewer reports to us, please specify this in the cover letter, mentioning the name of the previous journal and the manuscript ID the study was given, and include a point-by-point response to reviewers that details how you have or plan to address the reviewers' concerns. Please contact me at the email that can be found below my signature if you have questions. 

Please re-submit your manuscript within two working days, i.e. by Oct 17 2021 11:59PM.

Kind regards,

Paula

Paula Jauregui, PhD

Associate Editor

PLOS Biology

---

## [Decision Letter · Decision Letter 1]

7 Jan 2022

Dear Dr Doan,

Thank you for submitting your manuscript "Dynamic proton-dependent motors power Type IX secretion and gliding adhesin movement in Flavobacterium" for consideration as a Research Article at PLOS Biology. Your manuscript has been evaluated by the PLOS Biology editors, an Academic Editor with relevant expertise, and by two independent reviewers. We had recruited a third reviewer, but they have not returned comments in a timely fashion.

I have taken over handling your paper from my colleague Dr Paula Jauregui, who is now on maternity leave. You'll see that both of the reviewers are broadly positive about your study, but each raises a number of concerns, some of which may require additional data and/or analysis to address.

In light of the reviews (below), we are pleased to offer you the opportunity to address the comments from the reviewers in a revised version that we anticipate should not take you very long. We will then assess your revised manuscript and your response to the reviewers' comments and we may consult the reviewers again.

We expect to receive your revised manuscript within 1 month.

**IMPORTANT - SUBMITTING YOUR REVISION**

*Resubmission Checklist*

*Published Peer Review*

*PLOS Data Policy*

*Blot and Gel Data Policy*

Sincerely,

Roli Roberts

Roland G Roberts PhD

Senior Editor

PLOS Biology

rroberts@plos.org

REVIEWERS' COMMENTS:

Reviewer #1:

The manuscript by Vincent and co-authors builds on recent exciting developments understanding the structure and function of the T9SS motor in protein secretion and gliding motility within the Bacteroidetes phylum. Specifically this paper focuses on how PMF is converted by the GldLM inner membrane motor proteins into mechanical torque to the outer membrane apparatuses for secretion/motility. New findings include a GldL E59A substitution mutant that uncouples protein secretion from motility; NMR evidence for protonation of GldL glutamate residues within an IM channel involved in proton flux; uses genetic assay to map how TMHs in GldL & M interact; found dissipating the pH gradient with nigericin hinders motility; provided evidence that PMF influences GldM conformational state and that GldLM exist in stationary and mobile complexes in vivo. A number of experiments also confirmed prior results from the homologous P. gingivalis T9SS conducted by this group or results from other groups in related/identical systems. Overall the data fits well and builds upon other findings, including a recent report a by Hennell James et al, 2021, which reports on the structure and function of the GldLM proton-driven motor. In general, the experiments are clever, carefully done and the text is clearly written.

Specific comments:

Fig. S2B. The TMH region shown for GldM is different than the one demonstrated in Hennell James et al, 2021 (see Fig. 2d). Importantly, E31 in Hennell James et al resides outside of the TMH and may explain why the E31A mutant had no phenotype. This discrepancy needs to be addressed/justified. 

Fig. 6A. The average number of foci per cell for GldL is shown. Since similar immunofluorescent experiments were done with GldM, GldK and GldN, their average foci number per cell should also be shown.

In Fig. 6B experiments, is it possible to quantify the fraction of foci that are static vs mobile? Additionally, at later time points (e.g. 1 to 10 min) do stationary foci become mobile and vice versa? That is, do specific GldLM complexes permanently reside as 'static secretion machines' or 'motile motility motors'?

Please provide and explanation for why NBalfa-sfGFP labeling of GldL-alpha produced a functional reporter, while in contrast various GldL and GldM fluorescent protein fusions did not. In the former case a much larger reporter complex is made; therefore it is not obvious why the latter constructs were non-functional.

The results with GldL-alpha/NbAlfa-sfGFP are intriguing but some controls are absent. (i) For example, a western of GldL-alpha to test if it is processed, which may explain why some populations are mobile, while the majority are static. (ii) NbAlfa-sfGFP reporter is a noncovalent interaction with GldL-alpha and therefore dissociates. Again, this could explain why some foci are mobile. Given this is an atypical reporter, are there control proteins where the alpha peptide is fused to proteins that are known to be static and mobile? (iii) GldL-alpha is complexed with GldM, whose four extended domains span through the periplasm and cell wall. Therefore, there is an apparent conceptual issue for how GldLM could be mobile when it is predicted to be trapped within a peptidoglycan network. 

Proteins/complexes in Fig. 6F/legend are poorly labeled/described. For example, is the gray IM cylinder the Sec machinery? If so, please label. Second, in the red OM cylinder, SprA is not described, but should be present as it serves as the channel for protein secretion. Third, what are the red/gray ovals under the red OM cylinder? Fourth, the secretion of SprB (red) is confusing. In the periplasm, it appears folded, while outside the cell it appears unfolded. Finally, what is the black triangle in the LPS layer (top), and what is the green hexagon in the OM and the green hatch below it in the bottom/right cartoon? 

The significance of a sub-population of GldL (or GldLM) moving is unclear. To support or describe the "rack and pinion" model the GldLM motors could simply reside in fixed locations whereby they rotate/spin and hence move the "tread/track" that's presumably attached to the SprB cell surface adhesin.

A major weakness in the proposed "rack and pinion" model proposed by other authors and supported by these authors, is the lack of a known tread/track network required to translocate SprB on the cell surface. 

Lines 722-23: The Song et al reference lacks a journal name and lacks an actual citation in Google Scholar. Importantly, this reference includes authors from this group and appears to provide the first description/evidence for conformational changes in GldM, which is expanded upon in this work.

Movie S1 not provided.

Minor comments:

The OM-associated ring is described two ways. On lines 81-82 it consists of GldK and GldN, while lines 171 & 882-3 it includes GldO. Please clarify.

Lines 168-69: to aid the reader, please note here that the soluble domains of GldL and GldM reside in the cytoplasm and periplasm, respectively, so the finding that these domains do not interact is expected.

Line 169: Please provide reference for the PorKLMN complex discussion.

Line 302: Please describe where in the alfa-tag was introduced in-frame in GldL in the Result/Discussion section.

Lines 738-739. Kellenberger et al reference is listed twice (one of which is partial). Additionally, this reference is not cited in the text.

In Fig. 2B, to aid the reader, label the N and C ends of GldL/M.

Fig. 2C legend, line 786. For clarity after, "Measurements" insert "(transcriptional repression)." Also, lines 788-89, for clarity, after "Interactions with TssL1 (in gray) served as negative controls" add "or positive control for self-interactions."

Fig. 6A, right graph. Change the y-axis description to "Number of foci/cell" if that is the intent.

In Figs. 6B-E and S3B please state how long NBalpha-sfGFP was induced with IPTG prior to imaging.

Fig. 6D legend, line 870: What are "filled spots" and "empty spots?" I see blue and black circles (spots) in the line graph, but no empty spots.

Reviewer #2:

This is an interesting paper that describes the driving mechanism of the gliding motility and the type 9 secretion of Flavobacterium johnsoniae.

The gliding motility and the type 9 secretion are powered by PMF. In this study, the authors clearly indicated that gliding motility is powered by the delta pH component of PMF but not by the membrane potential component. An inner membrane complex, GldLM, is believed to be a proton channel and act as a motor to drive these processes. The authors demonstrated that glutamate residues (E49 and E59) in the transmembrane helix 2 of GldL are important for the GldLM function. Interestingly, E59 is essential for the gliding motility but not for type 9 secretion, whereas E49 is essential for type 9 secretion. Moreover, they revealed localization and movement of GldLM and SprB, and their data support the tread model.

These findings greatly contribute to our understanding of the mechanism of the gliding motility as well as the type 9 secretion.

However, I found some issues that should be clarified before publication.

(1) The result of limited proteolysis shown in Fig.3A seems inconsistent with that in Fig. 4H. Addition of CCCP (proton ionophore) changed the GldM structure to a protease sensitive form (Fig. 3A). However, the gldL E59A mutation, which may block the proton channel for gliding motility, did not change GldM to the protease sensitive form (Fig. 4H).

(2) Line 198-200 

This is not true. A kink between PorM D1 and D2 has been found in the crystal structure of the entire periplasmic region of PorM. "Hennell James, Deme et al" observed PorLM and did not show the D1-D2 kink of GldM in their CryoEM structure.

(3) Fig. 3B

The direction of the black arrow may be wrong.

(4) Line 254-

Did peptides form a complex like the GldL5+GldM2 complex in the micelles?

If not, I think it is nonsense to discuss the protonation of the glutamate.

(5) Line 722 "Song, Perpich et al. 2021".

Incomplete description. I could not find the reference paper.

(6) Fig.1

How much time do you needed to recover the motility after washing CCCP out?

---

## [Editor Report · Decision Letter 2]

21 Feb 2022

Dear Dr Doan,

Thank you for submitting your revised Research Article entitled "Dynamic proton-dependent motors power Type IX secretion and gliding adhesin movement in Flavobacterium" for publication in PLOS Biology. The Academic Editor and I have now assessed your responses and revisions. 

Based on this assessment, we will probably accept this manuscript for publication, provided you satisfactorily address the remaining points raised by the reviewers. Please also make sure to address the following data and other policy-related requests.

IMPORTANT:

a) Please change your title to "Dynamic proton-dependent motors power Type IX secretion and gliding motility in Flavobacterium"

b) Please address my Data Policy requests below; specifically, we need you to supply the numerical values underlying Figs 1B, 2CD, 4BDEG, 5B, 6D, S3B. Please also cite the location of the data clearly in each relevant main and supplementary Fig legend, e.g. “Data underlying this Figure can be found in S1 Data”.

We expect to receive your revised manuscript within two weeks. 

*Published Peer Review History*

*Early Version*

Sincerely,

Roli Roberts

Senior Editor,

rroberts@plos.org,

PLOS Biology

DATA POLICY:

Regardless of the method selected, please ensure that you provide the individual numerical values that underlie the summary data displayed in the following figure panels as they are essential for readers to assess your analysis and to reproduce it: Figs 1B, 2CD, 4BDEG, 5B, 6D, S3B. NOTE: the numerical data provided should include all replicates AND the way in which the plotted mean and errors were derived (it should not present only the mean/average values).

We require the original, uncropped and minimally adjusted images supporting all blot and gel results reported in an article's figures or Supporting Information files. We will require these files before a manuscript can be accepted so please prepare and upload them now. Please carefully read our guidelines for how to prepare and upload this data: https://journals.plos.org/plosbiology/s/figures#loc-blot-and-gel-reporting-requirements 

DATA NOT SHOWN?

---

## [Editor Report · Decision Letter 3]

1 Mar 2022

Dear Dr Doan,

On behalf of my colleagues and the Academic Editor, Michael Laub, I'm pleased to say that we can in principle accept your Research Article "Dynamic proton-dependent motors power Type IX secretion and gliding motility in Flavobacterium" for publication in PLOS Biology, provided you address any remaining formatting and reporting issues. These will be detailed in an email that will follow this letter and that you will usually receive within 2-3 business days, during which time no action is required from you. Please note that we will not be able to formally accept your manuscript and schedule it for publication until you have any requested changes.

PRESS: We frequently collaborate with press offices. If your institution or institutions have a press office, please notify them about your upcoming paper at this point, to enable them to help maximise its impact. If the press office is planning to promote your findings, we would be grateful if they could coordinate with biologypress@plos.org. If you have not yet opted out of the early version process, we ask that you notify us immediately of any press plans so that we may do so on your behalf.

Sincerely,

Roli Roberts

Roland G Roberts, PhD 

Senior Editor 

PLOS Biology

rroberts@plos.org